# Rapid detection and molecular epidemiology of β-lactamase producing *Enterobacteriaceae* isolated from food animals and in-contact humans in Nigeria

**Solomon Olabiyi Olorunleke**[1,2,3ʘ¤]*, **Miranda Kirchner**[1ʘ], **Nicholas Duggett**[1‡],
**Kim Stevens**[2‡], **Kennedy F. Chah**[4ʘ], **John A. Nwanta**[3‡], **Lucy A. Brunton**[2ʘ], **Muna
F. Anjum**[1ʘ]

**1** Department of Bacteriology, Animal and Plant Health Agency, Weybridge, Surrey, United Kingdom,
**2** Veterinary Epidemiology, Economics and Public Health Group, Department of Pathobiology and Population
Sciences, Royal Veterinary College, London, United Kingdom, **3** Department of Veterinary Public Health and
Preventive Medicine, University of Nigeria Nsukka, Enugu, Nigeria, **4** Department of Veterinary Pathology
and Microbiology, University of Nigeria Nsukka, Enugu, Nigeria

ʘ These authors contributed equally to this work.
¤ Current address: Department of Animal Science, Ebonyi State University, Abakaliki, Nigeria
‡ ND, KS and JAN also contributed equally to this work.
* solomon.olorunleke@ebsu.edu.ng

org/10.1371/journal.pone.0289190

Coast, GHANA

**Data Availability Statement:** All relevant data are
within the manuscript and its Supporting
Information files. All the whole genome sequences

## Abstract

The emergence and spread of β-lactamase-producing *Enterobacteriaceae* poses a significant threat to public health, necessitating the rapid detection and investigation of the molecular epidemiology of these pathogens. We modified a multiplex real-time (RT)-PCR to concurrently detect β-lactamase genes (*bla*$_{CTX-M}$, *bla*$_{TEM}$, and *bla*$_{SHV}$) and *Enterobacteriaceae* 16S ribosomal RNA. qPCR probes and primers were validated using control isolates, and the sensitivity and specificity assessed. The optimised multiplex qPCR was used to screen 220 non-clinical *Enterobacteriaceae* from food animals and in-contact humans in Southeast Nigeria selected on cefotaxime-supplemented agar plates. Binary logistic regression was used to explore factors associated with the presence of the *bla*$_{TEM}$ and *bla*$_{SHV}$ genes in these isolates, and a subset of isolates from matched sampling sites and host species were whole genome sequenced, and their antimicrobial resistance (AMR) and plasmid profiles determined. The sensitivity and specificity of the qPCR assay was 100%. All isolates (220/220) were positive for *Enterobacteriaceae* ribosomal 16S rRNA and *bla*$_{CTX-M}$, while 66.4% (146/220) and 9% (20/220) were positive for *bla*$_{TEM}$ and *bla*$_{SHV}$, respectively. The prevalence of *bla*$_{TEM}$ and *bla*$_{SHV}$ varied across different sampling sites (farm, animal market and abattoirs). Isolates from Abia state were more likely to harbour *bla*$_{TEM}$ (OR = 2.3, p = 0.04) and *bla*$_{SHV}$ (OR = 5.12,p = 0.01) than isolates from Ebonyi state; *bla*$_{TEM}$ was more likely to be detected in isolates from food animals than humans (OR = 2.34, p = 0.03), whereas the reverse was seen for *bla*$_{SHV}$ (OR = 7.23, p = 0.02). Furthermore, *Klebsiella* and *Enterobacter* isolates harboured more AMR genes than *Escherichia coli*, even though they were isolated from the same sample. We also identified pan resistant *Klebsiella* harbouring resistance to ten classes of antimicrobials and disinfectant. Therefore, we recommend

are available from the ENA database (study accession number: PRJEB63579).

**Funding:** The main funding for this study (Grant number: NGCN-2018-242) was provided by the Commonwealth Scholarship Commission in the UK (CSC), awarded to author SOO. Additional financial support was received from the Royal Veterinary College, University of London, awarded to author SOO, and the UK FAO Reference Centre for Antimicrobial Resistance, funded by the Department for Environment, Food and Rural Affairs (DEFRA) and the Department of Health and Social Care's Fleming Fund, awarded to author SOO. The funders played no role in study design, data collection and analysis, decision to publish, or manuscript preparation.

**Competing interests:** The authors have declared that no competing interests exist.

ESKAPE pathogens are included in AMR surveillance in future and suggest qPCRs be utilised for rapid screening of *Enterobacteriaceae* from human and animal sources.

## Introduction

Globally, about 1.27 million human deaths were accredited to AMR in 2019 and the death toll has been projected to reach 10 million deaths per year by 2050 if no global precautionary measures are actively taken [1, 2]. The highest proportion of the estimated number of deaths caused by AMR will be recorded in low-and-middle-income countries (LMICs) as infectious diseases still account for a greater proportion of preventable mortalities recorded in these countries [3]. AMR is a compound problem with a myriad of interconnected drivers which may include dynamic changes in human demographics, climate change, and trade [4], and concerted efforts are being made globally to slow down its spread.

The intensity of the burden of AMR in Africa is undermined due to the lack of accurate, robust data on phenotypic and genotypic AMR profiles from health and veterinary departments [5]. Bernabé et al. (2017) reported that the deficit of global response to AMR surveillance in West Africa is linked to the large gaps in data on the prevalence of AMR emanating from equipped laboratories and strategic surveillance networks being limited [6]. Globally, a group of bacteria known as ESKAPE (*Enterococcus spp*, *Staphylococcus aureus*, *Klebsiella pneumoniae*, *Acinetobacter baumanni*, *Pseudomonas aeruginosa*, and *Enterobacter spp*) pathogens have been identified to play a major role in the spread of multiple AMR genes [7], including resistance to third-generation cephalosporins. Also, production of extended-spectrum β-lactamase (ESBL) enzyme among *Enterobacteriaceae* is an important public health concern [8] as they acquire AMR plasmids or other mobile genetic elements and spread AMR genes among different epidemiological compartments [8, 9]. The global public health concern on increasing prevalence of AMR *Enterobacteriaceae* has prompted urgency in developing rapid and efficient diagnostic tools for prompt characterization [10]. The diversity of β-lactamase genes spreading globally has necessitated the need for development of a rapid screening test for their identification. Production of ESBLs is the most frequent mechanism *Enterobacteriaceae* employ to inactivate various β-lactam antibiotics. ESBLs predominantly consist of $bla_{CTX-M}$, $bla_{TEM}$, and $bla_{SHV}$, with a few point mutations at selected loci within the respective $bla_{TEM}$, and $bla_{SHV}$ genes giving rise to ESBL variants [11]. These enzymes are encoded by genes located on either the bacterial chromosomes or on plasmids, and so far over 1,300 β-lactamases have been described [12]. $bla_{CTX-M}$ has become the dominant ESBL with global prevalence [13, 14] and their spread has been attributed to its location on self-mobilizable non-host-specific plasmids [12].

[15] The conventional culture-based procedure for AMR testing has been widely used to phenotypically characterize ESBL producers. Although this method is laborious and time-consuming, it provides information for clinical management and surveillance. However, it does not provide information on the mechanism(s) of resistance for the manifested phenotype, and this serves as a major limitation of the culture-based method in AMR surveillance [12, 15, 16]. In clinical settings, the consequence of prolonged duration of AMR testing of *Enterobacteriaceae* could be deleterious as infection may get established, spread rapidly in hospital and community environments, and appropriate interventions could be delayed [12, 17].

Molecular methods can be more efficient than culture-based methods for detection of AMR and differentiation of bacterial pathogens [15]. However, methods such as whole

genome sequencing (WGS), which provides detailed characterisation of bacteria and is being used more frequently in surveillance [18, 19], remain expensive and out of reach for many countries. The development of multiplex real-time polymerase chain reaction (qPCR) has reduced the time required for AMR testing and can provide a rapid guided decision for clinicians, veterinarians or other end users [20]. Multiplex qPCR has made global AMR monitoring and surveillance easier as many samples can be processed within a short timeframe at a reduced reaction cost and with limited sample materials [16, 21]. Although several attempts have been made to develop multiplex qPCR for the detection of β-lactamase genes [11–13], there is still need for a high-throughput molecular technique to simultaneously identify *Enterobacteriaceae* and the β-lactamase genes they harbour. The aims of this study were: to develop and validate a multiplex qPCR for the simultaneous detection of ESBLs (CTX-M, TEM, and SHV) and *Enterobacteriaceae* ribosomal 16S rRNA gene; evaluate the ability of this assay to rapidly detect these AMR genes in *Enterobacteriaceae* isolated from livestock and humans with known phenotypic resistance; investigate factors (sampling levels, state and species) associated with presence of β-lactamase genes, and investigate the molecular epidemiology of AMR gene acquisition among *Enterobacteriaceae* isolated from the same host species and sample origin.

## Materials and methods

### Ethical statement

The methods/procedures used in this study were concomitant with that outlined in the Animals Scientific Procedures Act of 1986 for the care and use of animals for research purposes. Approval was obtained from the Research Ethics Committee of the Faculty of Veterinary Medicine, University of Nigeria, Nsukka (Approval Reference Number: FVM-UNN-IACUC-2019-0570).

### Selection and culturing of control isolates

Twenty-two control isolates harbouring different β-lactamase genes (S1 Table) previously described [22] were selected from the Animal and Plant Health Agency (APHA) frozen bead stock collection. The control strains were specifically chosen to serve as positive and negative controls in the detection of the various ESBL genes, including CTX-M (18 positives and 4 negatives), TEM (16 positives and 6 negatives), SHV (11 positives and 11 negatives), and 16s (18 positives and 4 negatives). These control strains were integral to the experimental design, facilitating the validation and reliability of the molecular assays conducted for the detection of respective genes.

### Selection of non-clinical *Enterobacteriaceae* isolates

Two hundred and twenty non-clinical *Enterobacteriaceae* isolates selected from food animals and in-contact human samples collected at the abattoir, animal market, and farms in Southeast Nigeria were characterized by PCR for $bla_{CTX-M}$, $bla_{TEM}$ and $bla_{SHV}$ (S2 Table). The criteria used for selection of the isolates were that they were phenotypically multi-drug resistant (MDR) and provided a representative example of the isolates from different sample origins (species, states, and location). The selected isolates purified on cefotaxime-supplemented Mac-Conkey agar comprised *Escherichia coli* (89%, 196/220), *Klebsiella pneumoniae* (8%, 17/220), and *Enterobacter cloacae* (3%, 7/220). PCR data for these genes from *E. coli* isolates have been described in Olorunleke et al. [8]

## Preparation of lysates

All isolates were cultured on CHROMagar ECC and a 10 μl loop full of bacteria was suspended into 200 μl of DNase-free water in an Eppendorf tube which was then placed in a preheated (99°C) ThermoFisher heat block for 15 minutes [23]. The supernatant containing DNA was subsequently used for qPCR.

## Validation of β-lactamase qPCR primers

A multiplex qPCR for β-lactamases ($bla_{TEM}$, $bla_{SHV}$, and $bla_{CTX-M}$) was performed with the selected control isolates to validate the qPCR primers and probes using the AriaMX Agilent Technologyshv. The methods [11] were modified by inclusion of an *Enterobacteriaceae* ribosomal 16S gene. The qPCR primers in this study targeted $bla_{TEM}$, $bla_{SHV}$, and $bla_{CTX-M}$ (CTX-M-group 1 and 9). The primers and probes (Sigma-Aldrich, Gillingham, UK) presented in S3 Table were reconstituted in nuclease-free water (Qiagen) before use and diluted to a concentration of 1μM for probes and 10 μM for primers. The qPCR reaction concentration, volumes, and cycling conditions were performed as described [11] (S4 and S5 Tables). At the end of the run (about 68 minutes), the cycle quantification (Cq) values were obtained at the point where each sample's reaction curve intersected the threshold line. The Cq values of each sample was used to determine presence or absence of the β-lactamase genes. This experiment was repeated seven times for each isolate in the control panel and the average Cq values were recorded for each isolate.

Likelihood of the assay revealing false-negative and false-positive results for the primers and probes of each AMR gene was evaluated by calculating the sensitivity and specificity of the assay using a 2 x 2 contingency table. Furthermore, the reliability and precision of the assay were also evaluated as described previously [24].

The results were used as a calibrated standard for the test isolates from food animals and in-contact humans in Southeast, Nigeria (n = 220). The Cq values obtained were compared to the range previously described for each of the validated assays. Those within the range were termed as positive (gene present) and those with no Cq values or greater than the range at the end of the cycle were termed as negative (gene absent) for the resistance genes tested.

## Whole genome sequencing of isolates from same host species and sampling site

Illumina NextSeq platform was used to whole genome sequence all the *K. pneumoniae* and *E. cloacae* isolates concurrently isolated with *E. coli* from the same host species and sampling site. The APHA SeqFinder pipeline was used to identify the AMR genes harboured by each isolate and Abricate was used to identify the plasmid replicons in the genomes assembled using SPAdes v3.12.0 [8, 25–27]. All the whole genome sequences were deposited at ENA (study accession number: PRJEB71718).

## Statistical analysis

Binary logistic regression was used to explore factors associated with presence of the β-lactamase genes ($bla_{SHV}$ and $bla_{TEM}$, dependent variables) among *Enterobacteriaceae* isolates. The independent variables assessed were animal species, sample origin, and state. These were first coded into dummy variables before analysis using SPSS software v26. The multivariable model was built using an automated forward stepwise approach taking forward for inclusion of any variable with a p-value $< 0.02$.

## Results

### Validation of ESBL positive controls using multiplex qPCR

The mean Cq values and standard error of seven technical repeat experiments for the multiplex qPCR ($bla_{CTX-M}$, $bla_{SHV}$, $bla_{TEM}$, and ribosomal 16S genes) are presented in Fig 1. The Cq values for isolates that were positive for the ribosomal 16S rDNA gene in *Enterobacteriaceae* ranged between 16.8 ± 0.5 to 20.4 ± 0.6. The non-*Enterobacteriaceae* isolates in the control panel (HPA 5, HPA 7, HPA 93, and HPA 96) had no Cq values for the 16S gene. The control isolates that were positive for $bla_{TEM}$ had a Cq value that ranged from 17.9 ± 0.2 to 21.6 ± 0.5 while those positive for $bla_{SHV}$ had a range between 15.6 ± 0.2 to 19.1 ± 0.1. When the threshold line of the reaction curve was adjusted, all the variants of $bla_{CTX-M}$ were detected by the assay and the positives ranged between 15.8 ± 0.5 to 25.9 ± 1.4. HPA 31 recorded the highest Cq value (25.9 ± 1.4) for the $bla_{CTX-M}$ gene which may be due to presence of $bla_{CTX-M-26}$ (CTX-M-25 group) in the isolate that was different from $bla_{CTX-M-15}$ (CTX-M-1 group) which was present in other isolates.

The calculated sensitivity and specificity of the assays were 100% respectively, while the reliability of the assay, as shown by the accuracy and precision values, was also 100%. The accuracy and precision values affirmed that this method could be used in future as a rapid tool for the detection of β-lactamases in *Enterobacteriaceae* using AriaMX Agilent technology. HPA70, 59, 46, and 31 were selected as positive controls for the multiplex qPCR ($bla_{CTX-M}$, $bla_{SHV}$, and $bla_{TEM}$) and B2318 as a negative control in the subsequent detection of β-lactamases by qPCR in the 220 test isolates.

### Prevalence of ESBL genes in the test isolates from food animals and in-contact humans in Southeast Nigeria

The prevalence of each of the β-lactamase genes among the 220 test isolates is presented in Table 1. All 220 isolates were positive for *Enterobacteriaceae* ribosomal 16S rDNA and $bla_{CTX-M}$ since they were selectively isolated from cefotaxime-supplemented MacConkey agar. Overall, 66.4% (146/220) and 9.0% (20/220) were positive for the $bla_{TEM}$ and $bla_{SHV}$, respectively.

The prevalence of $bla_{TEM}$ and $bla_{SHV}$ varied among species of *Enterobacteriaceae* isolated from different sampling levels (Table 2). The $bla_{SHV}$ gene was detected less frequently than the other β-lactamase genes screened. As $bla_{SHV}$ is intrinsic to *K. pneumoniae* it was detected in all *Klebsiella* isolates (100%, 17/17) but only in 2.0% (4/196) of the *E. coli*. At the sample level, prevalence of $bla_{TEM}$ among *Enterobacteriaceae* isolates varied from 53.1% to 71.4%, being lowest in chicken and highest in cattle isolates. The prevalence of $bla_{SHV}$ was highest in human isolates (22.9%, 8/35), and among food animals it was most prevalent in chicken isolates (15.6%, 5/32). None of the pig isolates were positive for $bla_{SHV}$. The prevalence of $bla_{SHV}$ among *E. coli* isolated from abattoir and animal market ranged between 3–5%, however, none of the isolates from the farm harboured $bla_{SHV}$. Likewise, none of the *E. cloacae* harboured $bla_{SHV}$. Furthermore, in this study, the prevalence of $bla_{SHV}$ among *Klebsiella spp* was the same regardless of the sampling site and host species.

Collectively, the prevalence of $bla_{TEM}$ and $bla_{SHV}$ was highest among isolates from abattoirs, and lowest among isolates from animal farms (Table 1). The highest prevalence of $bla_{TEM}$ was observed in the *K. pneumoniae* isolates (94.1%, 16/17) and the lowest was in *E. coli* (62.8%, 123/196). Prevalence of $bla_{TEM}$ among *E. coli* isolates varied as animals and in-contact humans moved from the farms (58%, 57/99) to animal markets (67%, 26/39) and abattoirs (69%, 40/58). However, among *K. pneumoniae*, abattoir isolates (75%, 3/4) had the lowest prevalence of

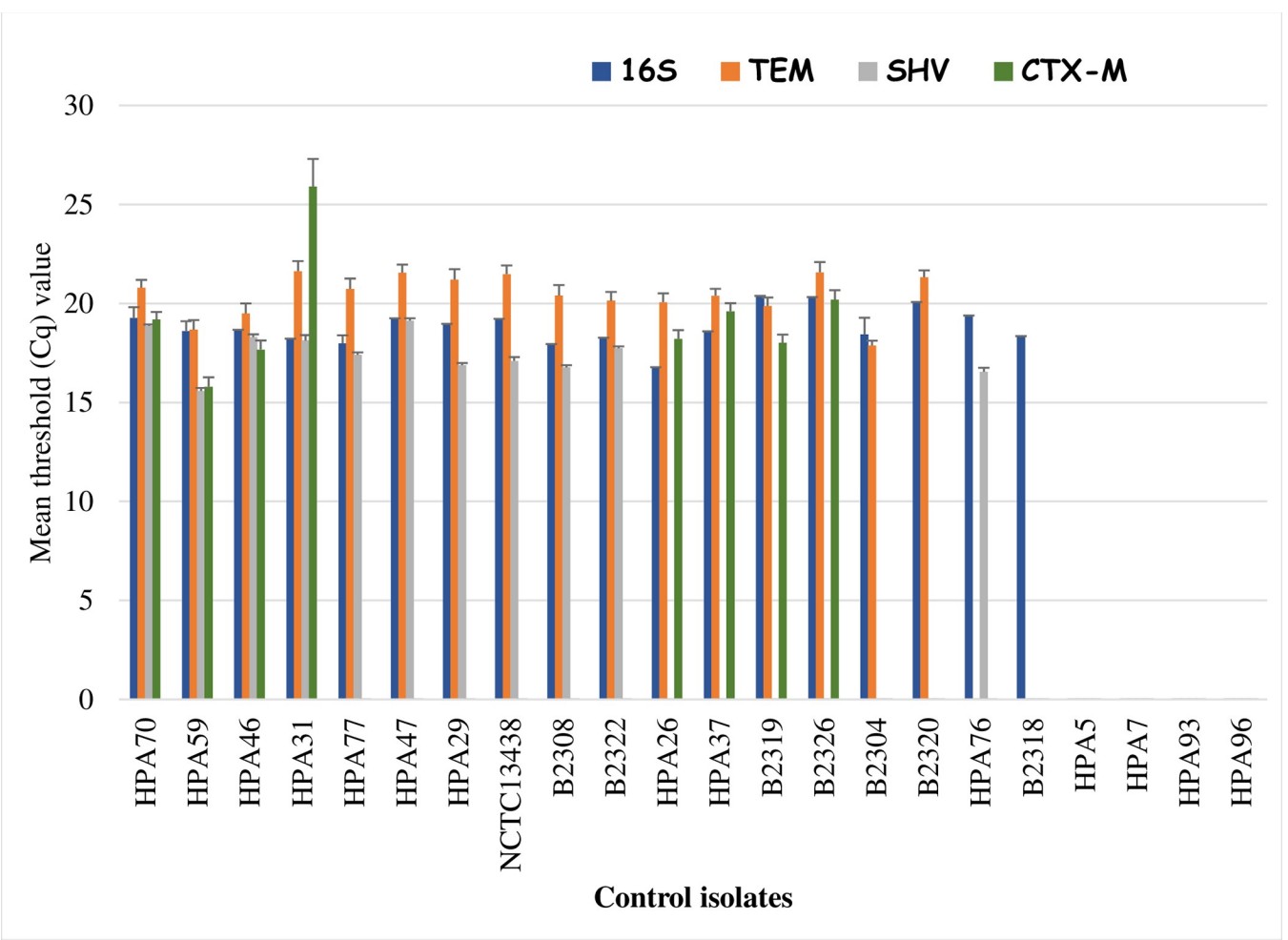

**Fig 1. Mean Cq values for 16S, TEM, SHV and CTX-M positive isolates.**

$bla_{TEM}$. Although no *E. cloacae* was isolated from farms in this study, the prevalence of $bla_{TEM}$ was highest among isolates from animal markets (100%, 3/3). At the host species level, the highest prevalence of $bla_{TEM}$ among *E. coli* was observed in isolates from cattle, and chicken isolates recorded the lowest prevalence. Among *E. cloacae*, the prevalence of $bla_{TEM}$ ranged between 50–100% in cattle, chicken and human host species. *E. cloacae* was not isolated from goat, pig or sheep. Similarly, *K. pneumoniae* was not isolated from pigs in this study, however, the prevalence of $bla_{TEM}$ ranged between 67–100% in other host species.

The distribution of β-lactamase genes varied among different states from which samples were obtained. Isolates obtained from Abia had a higher prevalence of $bla_{TEM}$ (76%) compared with Ebonyi and Enugu (62% and 64%, respectively). The prevalence of $bla_{SHV}$ was also highest among isolates obtained from Abia (16%) when compared with Ebonyi and Enugu (6% and 9%, respectively). The prevalence of $bla_{TEM}$ varied across the states in *E.coli* and *Enterobacter* species but was the same in *Klebsiella* isolates. Aside *Enterobacter* isolates that did not harbour $bla_{SHV}$ gene across the states, the prevalence of $bla_{SHV}$ among *E. coli* and *Klebsiella spp* isolated in Abia and Ebonyi were the same. The lowest prevalence of $bla_{SHV}$ was recorded in *E. coli* and *Klebsiella* isolated in Enugu State.

**Table 1. Prevalence of *Enterobacteriaceae* isolates from food animals and in-contact humans harbouring β-lactamase genes in Southeast Nigeria.**

| Variables | Prevalence ± Standard Error (%) | 95% Confidence Interval (%) |
|---|---|---|
| *TEM* | | |
| Isolate | | |
| *Escherichia coli* (n = 196) | 62.8 ± 3.5 | 55.9–69.6 |
| *Enterobacter* (n = 7) | 85.7 ± 2.3 | 57.9–100 |
| *Klebsiella* (n = 17) | 94.1 ± 6.3 | 80.4–100 |
| Animal species | | |
| Cattle (n = 42) | 71.43 ± 7.06 | 57.18–85.68 |
| Goat (n = 36) | 66.67 ± 7.97 | 50.49–82.84 |
| Sheep (n = 37) | 67.57 ± 7.80 | 51.74–83.39 |
| Chicken (n = 32) | 53.13 ± 8.96 | 34.85–71.40 |
| Pig (n = 37) | 64.86 ± 7.96 | 48.73–81.00 |
| Human (n = 36) | 69.44 ± 7.78 | 53.64–85.25 |
| Sample origin | | |
| Abattoir (n = 68) | 72.06 ± 5.52 | 58.99–81.01 |
| Animal Market (n = 46) | 71.74 ± 6.71 | 58.22–85.26 |
| Farm (n = 106) | 60.38 ± 4.82 | 51.03–70.13 |
| State | | |
| Abia (n = 50) | 76.00 ± 6.10 | 63.74–88.26 |
| Ebonyi (n = 106) | 62.26 ± 4.73 | 52.88–71.64 |
| Enugu (n = 64) | 64.06 ± 6.05 | 51.98–76.14 |
| *SHV* | | |
| Isolate | | |
| *Escherichia coli* (n = 196) | 2.04 ± 1.01 | 0.04–4.04 |
| *Enterobacter* (n = 8) | 0.00 | 0.00 |
| *Klebsiella* (n = 16) | 100% | - |
| Animal species | | |
| Cattle (n = 42) | 7.14 ± 4.02 | 0.98–15.27 |
| Goat (n = 36) | 5.56 ± 3.87 | 2.30–13.42 |
| Sheep (n = 37) | 5.41 ± 3.77 | 2.24–13.05 |
| Chicken (n = 32) | 15.63 ± 6.52 | 2.32–28.93 |
| Pig (n = 37) | 0.00 | 0.00 |
| Human (n = 36) | 22.22 ± 7.03 | 7.96–36.49 |
| Sample origin | | |
| Abattoir (n = 68) | 11.76 ± 3.83 | 3.79–19.07 |
| Animal Market (n = 46) | 10.87 ± 4.64 | 1.52–20.21 |
| Farm (n = 106) | 6.60 ± 2.47 | 1.83–11.63 |
| State | | |
| Abia (n = 50) | 16.00 ± 5.23 | 5.48–26.52 |
| Ebonyi (n = 106) | 5.66 ± 2.25 | 1.19–10.13 |
| Enugu (n = 64) | 9.38 ± 3.67 | 2.04–16.71 |

## Factors associated with TEM and SHV gene prevalence among *Enterobacteriaceae* isolates from food animals and in-contact humans in Southeast Nigeria

The results of the multivariable analysis for $bla_{TEM}$ and $bla_{SHV}$ are presented in Table 3. Isolates obtained from Abia were observed to be 2.30 (95% CI: 1.02–5.16, p = 0.04) times more

**Table 2. Comparative prevalence of TEM and SHV gene in *Escherichia coli*, *Enterobacter cloacae* and *Klebsiella pneumoniae* from different sampling levels.**

| Isolate | *Escherichia coli* | | *Enterobacter cloacae* | | *Klebsiella pneumoniae* | |
|---|---|---|---|---|---|---|
| | TEM | SHV | TEM | SHV | TEM | SHV |
| **Sampling Site** | | | | | | |
| Abattoir | 40/58 (69%) | 2/58 (3%) | 3/4 (75%) | | 5/6 (83%) | 6/6 (100%) |
| Animal Market | 26/39 (67%) | 2/39 (5%) | 3/3 (100%) | | 4/4 (100%) | 4/4 (100%) |
| Farm | 57/99 (58%) | | | | 7/7 (100%) | 7/7 (100%) |
| **Host Species** | | | | | | |
| Cattle | 27/37 (73%) | | 1/2 (50%) | | 2/3 (67%) | 3/3 (100%) |
| Chicken | 12/28 (43%) | | 1/1 (100%) | | 4/4 (100%) | 4/4 (100%) |
| Goat | 23/35 (66%) | 2/35 (6%) | | | 1/1(100%) | 1/1 (100%) |
| Human | 13/23 (57%) | 1/23 (4%) | 4/4 (100%) | | 8/8 (100%) | 8/8 (100%) |
| Pig | 24/37 (65%) | | | | | |
| Sheep | 24/36 (67%) | 1/36 (3%) | | | 1/1 (100%) | 1/1 (100%) |
| **State** | | | | | | |
| **Abia** | 30/40 (75%) | 1/40 (3%) | 3/4 (75%) | | 5/5 (100%) | 5/5 (100%) |
| **Ebonyi** | 58/98 (59%) | 3/98 (3%) | 3/3 (100%) | | 5/5 (100%) | 5/5 (100%) |
| **Enugu** | 35/58 (60%) | | | | 7/7 (100%) | 6/7 (86%) |

likely to harbour a $bla_{TEM}$ gene than those from Ebonyi State. Overall, there was no significant difference in the odds of $bla_{TEM}$ being present across sampling levels and species, however, they were retained in the model because including them influenced the estimates for State. It was observed that isolates from food animals were 2.34 (95% CI: 1.10–4.98, p = 0.03) times more likely to harbour $bla_{TEM}$ than human isolates.

Isolates from Abia state were 5.12 (95% CI: 1.42–18.37, p = 0.01) times more likely to harbour a $bla_{SHV}$ gene than those isolated from Ebonyi state. The odds of the $bla_{SHV}$ gene being present were reduced in isolates from cattle (OR: 0.15, 95% CI: 0.03–0.75, p = 0.02) and goats (OR: 0.18, 95% CI: 0.03–1.01, p = 0.05) compared to humans. There was no significant difference in the odds of an isolate harbouring the $bla_{SHV}$ gene across sampling levels. However, this variable was retained in the model because including it influenced the estimates for the other variables. Furthermore, it was observed that *Enterobacteriaceae* isolated from in-contact humans were 7.23 (95% CI: 1.33–39.23, p = 0.02) more times likely to harbour $bla_{SHV}$ than isolates from animals.

## Molecular epidemiology of AMR gene acquisition among *Enterobacteriaceae* isolated from the same host species and sample origin

Using WGS, we determined that the mean number of AMR genes harboured by *E. coli*, *E. cloacae* and *K. pneumoniae* isolated from food animals and in-contact humans in this study were 8, 12, and 15, respectively (Fig 2). The numbers of AMR genes and plasmids varied between *E. coli*, *E. cloacae* and *K. pneumoniae* even when isolates were purified from the same sample site and host species (S6 Table). For instance, in one goat farm (Farm A) the total number of AMR genes harboured by *E. coli* isolates varied between 3 and 16. *K. pneumoniae* isolates from the same farm harboured 12 genes. Generally, we observed that all the *Klebsiella* isolates predominantly harboured different variants of IncF replicon bearing plasmids in addition to other plasmid replicon types (IncN, IncY, IncH, IncX, IncR, Col, and ColRNAI). Furthermore, two *K. pneumoniae* ST-39 isolated from chicken host species at an animal market harboured 26 AMR genes while the *E. coli* isolated from the same site and host species harboured only 9 AMR genes.

**Table 3. Factors associated with TEM and SHV gene prevalence among *Enterobacteriaceae* isolates from food animals and in-contact humans in Southeast Nigeria.**

| β-Lactamase genes (Independent variables) | Predictors Variable | Explanatory variables | Odds ratio | 95% Confidence Interval | P-value |
|---|---|---|---|---|---|
| TEM | Species | Human | Reference variable | | |
| | | Cattle | 1.06 | 0.39–2.91 | 0.91 |
| | | Chicken | 0.52 | 0.18–1.50 | 0.22 |
| | | Goat | 0.82 | 0.30–2.25 | 0.70 |
| | | Pig | 0.83 | 0.28–2.46 | 0.74 |
| | | Sheep | 1.03 | 0.37–2.85 | 0.95 |
| | Sample origin | Abattoir | 1.72 | 0.76–3.91 | 0.19 |
| | | Market | 1.30 | 0.62–2.73 | 0.49 |
| | | Farm | Reference variable | | |
| | State | **Abia** | **2.30** | **1.02–5.16** | **0.04** |
| | | Enugu | 1.15 | 0.57–2.32 | 0.69 |
| | | Ebonyi | Reference variable | | |
| | Human/Animal | Human | Reference variable | | |
| | | **Animal** | **2.34** | **1.10–4.98** | **0.03** |
| SHV | Species | Human | Reference variable | | |
| | | **Cattle** | **0.15** | **0.03–0.75** | **0.02** |
| | | Chicken | 0.63 | 0.15–2.60 | 0.52 |
| | | **Goat** | **0.18** | **0.03–1.01** | **0.05** |
| | | Pig | Omitted* | | |
| | | Sheep | 0.26 | 0.05–1.46 | 0.13 |
| | Sample origin | Abattoir | 1.79 | 0.49–6.62 | 0.38 |
| | | Market | 2.31 | 0.58–9.18 | 0.24 |
| | | Farm | Reference variable | | |
| | State | **Abia** | **5.12** | **1.42–18.37** | **0.01** |
| | | Enugu | 3.30 | 0.93–11.69 | 0.07 |
| | | Ebonyi | Reference variable | | |
| | Human/Animal | **Human** | **7.23** | **1.33–39.23** | **0.02** |
| | | Animal | Reference variable | | |

* No positive isolate was recorded

In this study, *K. pneumoniae* and *E. cloacae* concurrently isolated with *E. coli* from the same abattoir samples usually harboured more resistance genes (S6 Table). The predominant β-lactamases harboured by *K. pneumonia* and *E. cloacae* include variants of CTX-M, SHV and TEM. Regardless of the sequence type, *K. pneumoniae* harboured $bla_{\text{CTX-M-15}}$. In contrast, different $bla_{\text{SHV}}$ alleles were detected with ST-34, ST-1552, ST-39 harbouring $bla_{\text{SHV-26}}$, $bla_{\text{SHV-62}}$, and $bla_{\text{SHV-4}}$ respectively. We also identified other β-lactamase allelic variants than previously reported in *K. pneumoniae*, and *E. coli* in Nigeria among the *E. cloacae* isolates which includes CMH-1, ACT-16, -17, -25, and OXA-272. Aside from β-lactamase genes, *K. pneumoniae* and *E. cloacae* isolates in this study also harboured genes conferring resistance to aminoglycosides, macrolides, disinfectant, sulphonamide, trimethoprim, tetracycline, phenicol, fosfomycin, fluoroquinolone, and quinolone (S6 Table).

## Discussion

The critical priority accorded to ESBL-producing *Enterobacteriaceae* has necessitated the need for development of rapid and affordable detection methods for identifying AMR genes that

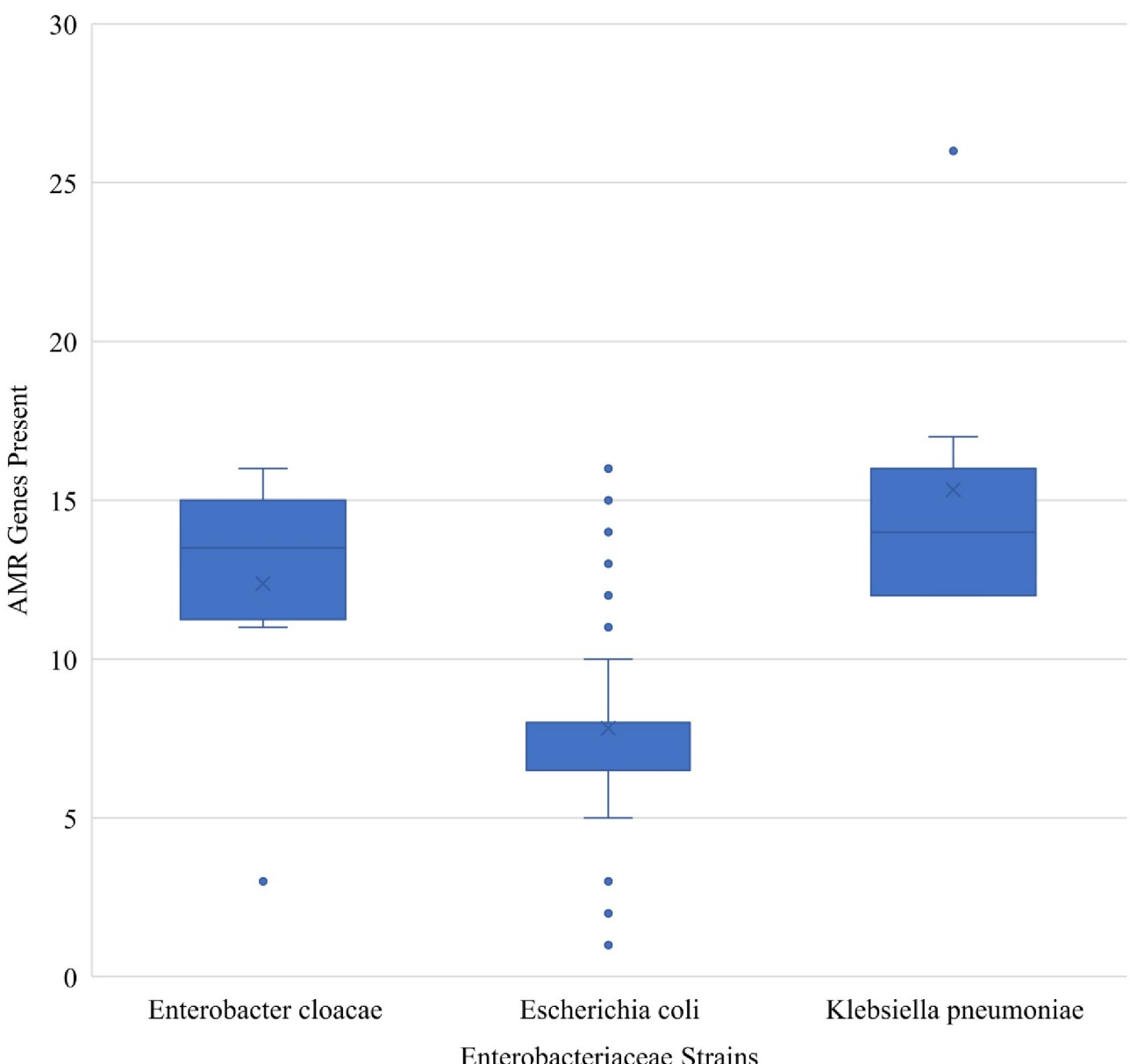

**Fig 2. Boxplot showing the total number of antimicrobial resistance genes present in each of the *Enterobacter cloacae* (n = 7), *Escherichia coli* (n = 196) and *Klebsiella pneumoniae* (n = 17) isolated from food animals and in-contact humans.**

confer resistance in *Enterobacteriaceae* isolated from clinical and community settings [28]. In this study, we modified a PCR-based method for the rapid detection of *Enterobacteriaceae* harbouring β-lactamase genes. Although several multiplex qPCR methods have been described for the detection of β-lactamases in *Enterobacteriaceae* [11, 13, 29], our assay did not only have an average turnaround time of 68 minutes but also incorporated the 16S ribosomal RNA gene specific for the *Enterobacteriaceae* family. The 16S ribosomal RNA gene is a highly conserved gene within bacterial species and among species of the same genus, but this gene is absent in eukaryotes and so can be used for rapid speciation of *Enterobacteriaceae* [30, 31]. In this study, we determined the mean Cq value (18.8) for the ribosomal 16S gene assay using the AriaMX

Agilent technology, which can be used as a standard for identifying *Enterobacteriaceae* in real-time with 100% sensitivity and specificity. The mean Cq values obtained for ESBL-positive isolates in this study were similar to those obtained by Roschanski et al. (2014) who used BioRad CFX96 and reported an average Ct value of 17.8, 17.2 and 20.1 for $bla_{TEM}$, $bla_{SHV}$ and $bla_{CTX-M}$ genes respectively. The highest average Cq value for $bla_{CTX-M}$ was recorded in HPA31 (25.9 ± 1.40) harbouring $bla_{CTX-M-26}$ which shares 99% amino acid identity with $bla_{CTX-M-25}$ but differs by three amino acids [32, 33]. Although the primer and probe were designed to identify CTX-M-1 and–M–9 groups, adjusting the threshold curve aided the detection of CTX-M-28 belonging to CTX-M-group 25 [11]. We noted that irrespective of the variants of $bla_{TEM}$, $bla_{SHV}$, and $bla_{CTX-M}$ harboured, as revealed by the WGS data, qPCR was able to detect any $bla_{TEM}$, $bla_{SHV}$, and $bla_{CTX-M}$. The sensitivity and specificity of each target gene in this study were 100% and this was similar to the report of Roschanski et al. (2014) but higher than the sensitivity (95%) and specificity (96.7%) of the commercial multiplex tandem PCR [34].

As all isolates were selectively obtained from cefotaxime-supplemented MacConkey agar, the presence of $bla_{CTX-M}$ was unsurprising and similar to that previously published [35, 36], although, in other studies, MacConkey agar was supplemented with ≤ 2μg/ml of cefotaxime [26] while in this study 5μg/ml was used. We observed in this study that the highest proportion of isolates harbouring $bla_{TEM}$ (93.8%) and $bla_{SHV}$ (100%) was *K. pneumoniae*. The latter is not surprising as $bla_{SHV}$ is intrinsic in *Klebsiella spp*.

Although animal species were not significantly associated with presence of $bla_{TEM}$, logistic regression analysis indicated, prevalence of $bla_{TEM}$ varied among different species. We noted *Enterobacteriaceae* isolated from livestock that have more contact with humans had the lowest $bla_{TEM}$ prevalence (e.g. chicken 51.5%). Aside from chicken isolates, *Enterobacteriaceae* isolated from livestock/ food animals were more likely to harbour a $bla_{TEM}$ gene than human isolates. However, isolates from food animals were less likely to harbour a $bla_{SHV}$ gene, and the majority of $bla_{SHV}$ containing isolates were from humans. Transmission between species and direction of transmission cannot be inferred from our data, but these findings suggest that both humans and animals are reservoirs of AMR [37]. The highest prevalence of $bla_{TEM}$ and $bla_{SHV}$ was observed among isolates from Abia and may be because Abia state is a business hub that attracts an influx of people from different parts of the country daily, consequently, increasing the demand for animal protein with influx of many animals. Although the predominance of $bla_{TEM}$ and $bla_{SHV}$ in specific host species have not been reported, our statistical analysis gave insight into the predominance of $bla_{TEM}$ among isolates from food animals and $bla_{SHV}$ among human isolates.

Although *E. coli* were more frequently recovered from samples and are considered a key organism in monitoring antimicrobial resistance spread in the food chain, the role of other *Enterobacteriaceae* like *K. pneumoniae* and *E. cloacae* in the dissemination of AMR genes should not be underestimated [8]. *Klebsiella pneumoniae* and *E. cloacae* harboured more antimicrobial resistance genes than *E. coli* isolated from the same sample origin and host species in this study. *Klebsiella pneumoniae* has been previously described as a key trafficker of MDR especially from the environment to clinically important bacteria [38]. Although *K. pneumoniae* are intrinsically resistant to ampicillin due to the presence of SHV-1 penicillinase in their chromosome [38], we observed in our data set that the sequence type determined which variant of SHV isolates harboured. Due to the acquisition of large conjugative plasmids, *K. pneumoniae* can harbour multiple AMR genes [39]. The IncF plasmids were the predominant plasmids in this study and are the most frequently described plasmid type from humans and animal sources [39, 40]. Also, we identified two pan-resistant isolates harbouring 4–6 plasmid replicons and up to 26 AMR genes conferring resistance to 10 different classes of antimicrobials and disinfectant. Although the AMR genes identified among *E. cloacae* isolates were similar to *K. pneumoniae* in this study, the predominant plasmid replicon in *E. cloacae* was Col, with the

exception of one found to harbour 6 plasmid replicons and a variety of β-lactamase genes including CMH-1. CMH-1 is a class C β-lactamase gene previously identified in *E. cloacae* isolated predominantly from human host [41, 42], and to the best of our knowledge, this is the first report of CMH-1 in food animals, suggesting its ecological niche is expanding. Also, just as our results indicate consideration of AMR in *Klebsiella* and *Enterobacter* species is an important aspect, in future, we should also consider virulence genes present in *E. coli* collected from ruminants, as they may harbour genes linked to human infection [43].

In conclusion, the continuous interaction between humans and animals may predicate the spread and persistence of AMR genes among food animals and in-contact humans. We established that the PCR-based methods described in this study could be used to identify three β-lactamase genes ($bla_{CTX-M}$, $bla_{TEM}$, and $bla_{SHV}$) accurately and rapidly, irrespective of variants harboured to monitor dissemination in *Enterobacteriaceae* isolated from human and food animals. Our results also suggest that by expanding current AMR surveillance programmes to include *Klebsiella* and *Enterobacter* spp would be beneficial as they are important reservoirs of AMR genes and harbour multiple plasmids for their horizontal transfer.

## Study limitation

The study acknowledges certain limitations pertaining to the sample size and the restricted number of *Klebsiella* and *Enterobacter spp* tested. While the study revealed a significant association between the geographical origin (States) of samples and the presence of β-lactamase genes, it is important to note that there may exist additional confounding factors not captured in the current study. These unaccounted variables could potentially exert an influence on the prevalence of $bla_{TEM}$ and $bla_{SHV}$ genes in Southeast Nigeria. Consequently, the study's findings are subject to variation, and a more comprehensive understanding may be attained through future research endeavours employing a larger and more diverse sample size.

## Supporting information

**S1 Table. AMR genotype of each isolate in the control panel.**
(DOCX)

**S2 Table. Enterobacteriaceae isolated from food animals and incontact humans in Southeast Nigeria.**
(XLSX)

**S3 Table. Primer and probe sequence.**
(DOCX)

**S4 Table. β-lactamase RT- PCR reaction mixture.**
(DOCX)

**S5 Table. RT-PCR conditions for 16S, TEM, SHV, and CTX-M reaction.**
(DOCX)

**S6 Table. AMR profiles of *Escherichia coli*, *Enterobacter cloacae* and *Klebsiella pneumoniae* isolated from the same sampling site and host species.**
(XLSX)

## Acknowledgments

We would like to express our sincere gratitude to the Animal and Plant Health Agency in the UK for generously providing their facility for this study.

## Author Contributions

**Conceptualization:** Solomon Olabiyi Olorunleke, Kennedy F. Chah, Lucy A. Brunton, Muna F. Anjum.

**Data curation:** Solomon Olabiyi Olorunleke, Nicholas Duggett, Kim Stevens, Lucy A. Brunton.

**Formal analysis:** Solomon Olabiyi Olorunleke, Nicholas Duggett, Kim Stevens, Kennedy F. Chah, Lucy A. Brunton, Muna F. Anjum.

**Funding acquisition:** Solomon Olabiyi Olorunleke, Lucy A. Brunton.

**Investigation:** Solomon Olabiyi Olorunleke, Miranda Kirchner, Kennedy F. Chah, Lucy A. Brunton, Muna F. Anjum.

**Methodology:** Solomon Olabiyi Olorunleke, Miranda Kirchner, Kennedy F. Chah, Lucy A. Brunton, Muna F. Anjum.

**Project administration:** Solomon Olabiyi Olorunleke, Miranda Kirchner, Kim Stevens, John A. Nwanta, Lucy A. Brunton, Muna F. Anjum.

**Resources:** Solomon Olabiyi Olorunleke, John A. Nwanta, Muna F. Anjum.

**Software:** Miranda Kirchner.

**Supervision:** Miranda Kirchner, Kennedy F. Chah, John A. Nwanta, Lucy A. Brunton, Muna F. Anjum.

**Validation:** Miranda Kirchner, Kennedy F. Chah, Lucy A. Brunton, Muna F. Anjum.

**Visualization:** Miranda Kirchner, Nicholas Duggett, Kennedy F. Chah.

**Writing – original draft:** Solomon Olabiyi Olorunleke.

**Writing – review & editing:** Solomon Olabiyi Olorunleke, Miranda Kirchner, Nicholas Duggett, Kim Stevens, Kennedy F. Chah, John A. Nwanta, Lucy A. Brunton, Muna F. Anjum.

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
