## [Decision Letter · Decision Letter 0]

23 Oct 2023

PONE-D-23-21745Rapid detection and molecular epidemiology of β-lactamase producing Enterobacteriaceae isolated from food animals and in-contact humans in Nigeria.PLOS ONE

Dear Dr. Olorunleke, 

Thank you for submitting your manuscript to PLOS ONE. After careful consideration, we feel that it has merit but does not fully meet PLOS ONE’s publication criteria as it currently stands. Therefore, we invite you to submit a revised version of the manuscript that addresses the points raised during the review process.

We look forward to receiving your revised manuscript.

Kind regards,

Kwame Kumi Asare, Ph.D

Academic Editor

PLOS ONE

Journal Requirements:

Reviewers' comments:

Reviewer's Responses to Questions

**Comments to the Author**

1. Is the manuscript technically sound, and do the data support the conclusions?

Reviewer #1: Yes

Reviewer #2: Yes

Reviewer #3: Yes

Reviewer #4: Yes

2. Has the statistical analysis been performed appropriately and rigorously? 

Reviewer #1: Yes

Reviewer #2: Yes

Reviewer #3: Yes

Reviewer #4: Yes

3. Have the authors made all data underlying the findings in their manuscript fully available?

Reviewer #1: Yes

Reviewer #2: Yes

Reviewer #3: Yes

Reviewer #4: Yes

4. Is the manuscript presented in an intelligible fashion and written in standard English?

Reviewer #1: Yes

Reviewer #2: Yes

Reviewer #3: Yes

Reviewer #4: Yes

5. Review Comments to the Author

Reviewer #1: Good work done, well written and described manuscript. This study will definitely contribute to knowledge in often low resources environment. Such method as describe here can definitely be used for surveillance with lower turnaround time.

Reviewer #2: This article presents a modified a multiplex real-time (RT)-PCR that can concurrently detect β-lactamase genes (blaCTX-M, blaTEM, and blaSHV) and Enterobacteriaceae 16S ribosomal RNA and provides useful data on the molecular epidemiology of AMR gene acquisition among Enterobacteriaceae

This study has prospect to be useful to the international AMR community, however there are some major deficiencies that needs to be addressed.

- The use of the acronym “RT-PCR” can be misleading as this is commonly used to represent reverse transcription PCR. Use of “qPCR” will be more appropriate.

- The study conclusion and recommendation seemed to be detached from the main study as the ESKAPE pathogens was not the main focus of this study, but Enterobacteriaceae of which only represent 3 out of the 6 ESKAPE pathogens. Moreover, the recommendation that ESKAPE pathogens be included in surveillance is strange as they are already identified as priority pathogens for AMR surveillance (at least in humans). I suggest conclusion be based on study findings and implication

- The interpretation of sensitivity and specificity should be done with caution considering the limited number of control strains and test isolate tested in this study. Especially the limited number of negative controls used in the control strain (4) and the test strains were selected based on their being able to grow in cefotaxime supplemented plates reducing the including of non ESBL producing strains

- The authors need to provide justification for focusing on the 3 ESBL genes with references to back it up.

- Figure 1 shows 22 control strains as opposed to 21 control strain which was indicated in the method.

- Table 1 adds little or no value to this article since all the reported parameters are 100%. It can be replaced with a sentence indicating 100% sensitivity, specificity, accuracy and precision.

- There is need to include well-articulated study limitations for example limited sample size, limited number of Klebsiella and Enterobacter tested etc

- The discussion in the current state seems to be quite repetitive of study findings. The authors should focus on discussing the key findings and their implication.

Other minor revisions include.

Pg 4 Line 85- “However, methods such as whole genome sequencing…….” This sentence is unwinding and needs to be rephrased.

Page 4 Line 74- I would use “affordable” rather than “cheap.”

Pg 5 Line 98: “evaluate the ability of this assay to rapidly detect these AMR genes in Enterobacteriaceae” I am a bit reserved about the use of “rapid”. PCR based test can hardly be regarded as rapid. Maybe when compared to traditional culture method but still constrained for point-of-care.

Pg 5 Line 106-107: Kindly provide more information on the control strains used in this section for example, how many positive and negative for ESBL of choice. More information about the source of control strains APHA should be provided in this section e.g. location

Pg 7 Line 151: “R gene”??? Do you mean resistant genes. Please provide initial definition

Pg 7 Line 159: Please write WGS in full since referring to “whole genome sequence” and not “whole genome sequencing.”

Pg 8 Line 168: “forward for inclusion any variable…”. Missing the word “of”

Pg 12 Line 227: The use of “as expected” is not necessary and should be expunged. Also quite repetitive across the manuscript

Pg 13 Table 3: Any reason why state level distribution was excluded from this table?

Pg 19 Line 336-346: This whole paragraph is a revision of literature and should be taken to the introduction section. The discussion section should focus on the implication of the study findings.

Pg 21 Line 383-386: “Although in this study, the state from which samples……” Take to study limitation.

Pg 22 Line 420: The sentence seems to be truncated.

Reviewer #3: REVIEWER’S COMMENTS FOR MANUSCRIPT PONE-D-23-21745

The authors purportedly developed a “rapid” RT-PCR assay for the detection of beta-lactamases in Enterobacteriaceae isolated from animal sources and in-contact humans in southeastern part of Nigeria. This is an important area of research given the global menace of antimicrobial resistance and the challenges this has posed to human health. The authors set out with four objectives in mind viz: 1. To develop and validate a multiplex RT-PCR for detection of ESBLs and Enterobacterial 16S rRNA gene

2. Evaluate the ability of the assay to “rapidly” detect the respective genes

3. Investigate factors associated with the presence of beta-lactamase genes

4. Investigate the molecular epidemiology of AMR gene acquisition among Enterobacteriaceae isolated from the same host species and sample origin.

However, while the present manuscript addressed objectives 1, 3 and 4, I could not say the same for objective 3. I could not see anywhere in the current manuscript where the authors made an attempt to evaluate the ability of the assay they developed in the study to “rapidly” detect the target genes. Interestingly, this is a keyword in the title of the manuscript. So, an important research question remains unanswered. How rapidly did this method detected the target genes compared with other methods that have been in existence?

Other minor comments are as below

Line 32: change to cefotaxime-supplemented agar plates

Lines 67-68: Authors should please follow the accepted scientific notation for writing spp. in scientific names. This should be done throughout the manuscript

Line 130: Author should please be specific about the function of AriaMX Agilent technology as a qPCR system

Line 219: Authors should please provide a reference to back up the assertion that blaSHV is intrinsic to K. pneumoniae

Lines 336-338 is more or less a repetition of lines 338-339

Lines 419-420: Incomplete sentence

Reviewer #4: Title: Rapid detection and molecular epidemiology of β-lactamase producing Enterobacteriaceae isolated from food animals and in-contact humans in Nigeria.

Authors: Olorunleke et al.

General Comments

The article is well written and seems to be the first to conduct in-depth studies on Enterobacteriaceae in three states in the South East of Nigeria. The report of pan resistance in some of these states is quite alarming also. However, there are a few grey areas that need to be explained.

Specific comments

Is it possible to add the cost of multiplex RT-PCR as against the cost of culture in our African setting where qPCR although on the increase in terms of usage due to covid, might not be popular for AMR?

In the supplementary Tables 2 & 6, genus and species names were not italicized.

I was looking out for the common genes that are shared in the states and which ones are found in the three states. In addition, I was looking forward to see at a glance the human isolates according to states and sites and the genes that are similar in each state since human isolates are 34 in number. This table should show any possible similarity with the animal species. I mean the AMR Table not plasmid table. I know there are comprehensive tables as supplementary files, but I would like to see at a glance a table with genes that are common and cut across the states.

Page 22 line 420, has an incomplete sentence, Lines 421 in page 22 to 423 in page 23, needs to be re-casted.

Lines 387 – 390, authors attributed highest prevalence of blaTEM 388 and blaSHV in Abia to be due to its commercial hub, while this may be partly true, it can also be seen from their results that Abia state had the lowest number of samples and so this could also be another attributable factor.

6. PLOS authors have the option to publish the peer review history of their article (what does this mean?). If published, this will include your full peer review and any attached files.

Reviewer #1: **Yes: **Babafela Awosile

Reviewer #2: **Yes: **Emelda Chukwu

Reviewer #3: No

Reviewer #4: No

---

## [Author Response · Author response to Decision Letter 0]

11 Dec 2023

Reviewer 1:

We sincerely appreciate your positive feedback and constructive comments on our manuscript, your encouraging words are both motivating and affirming, and we are grateful for the recognition of the potential impact of our study in resource-constrained environments. We are particularly pleased that you found the manuscript to be well-written and described. Your acknowledgement of the potential contribution to knowledge in low-resource settings is a testament to our commitment to addressing challenges faced in such environments. We share your vision that the methods outlined in our study can be instrumental in establishing efficient surveillance practices with reduced turnaround time.

Reviewer 2:

We would like to express our sincere gratitude for your thoughtful and comprehensive feedback on our manuscript. Your detailed comments and insights have been instrumental in refining our work, and we highly appreciate the time and effort you invested in the review process.

Addressing your concerns and suggestions, we have implemented the following revisions:

• Acronym Usage (RT-PCR): We appreciate your clarification regarding the use of "RT-PCR." We have revised the manuscript to replace it with "qPCR" to accurately reflect our methodology.

• Focus on ESKAPE Pathogens: We acknowledge your point about the focus on Enterobacteriaceae rather than all ESKAPE pathogens. We have adjusted the conclusion and recommendations to better align with the main study findings, ensuring a more cohesive narrative.

• Sensitivity and Specificity Interpretation: Sensitivity and specificity validation for the qPCR primers and probes was conducted using control strains rather than test isolates. Contrary to the impression given, the number of negative controls exceeded four, and the selection of control strains was meticulous. These strains served as both positive and negative controls for the detection of various ESBL genes, namely CTX-M (18 positives and 4 negatives), TEM (16 positives and 6 negatives), SHV (11 positives and 11 negatives), and 16s (18 positives and 4 negatives). It is crucial to clarify that the four isolates utilized as negative controls in the test isolate experiment underwent validation only after confirming the sensitivity and specificity of the primers and probes for detecting ESBLs. 

• Justification for ESBL Genes Focus: We have included a brief justification for focusing on the three ESBL genes in the introduction (Lines 70 – 82), supported by relevant references to provide context for our selection.

• Figure 1 and Table 1 Corrections: We have corrected the discrepancies in Figure 1 and addressed your suggestion to streamline Table 1 by providing a concise statement indicating 100% sensitivity, specificity, accuracy, and precision.

• Inclusion of Study Limitations: We have added a section specifically addressing the limitations of our study, including the limited sample size and the restricted number of Klebsiella and Enterobacter strains tested.

• Repetitive Sections and Unwieldy Phrases: We have revised and rephrased sections that were identified as repetitive or unwieldy, following your suggestions.

• Minor Revisions: We have addressed the minor revisions you pointed out, including clarifications on terms, grammar improvements, and adjustments to the discussion section.

Reviewer 3:

We acknowledge the concerns raised by Reviewer 3 regarding the evaluation of the assay's rapidity in detecting target genes, we acknowledge the oversight in explicitly addressing this aspect in the manuscript. We agree that it is a crucial point, especially considering its inclusion in the title. We have included a statement clearly describing the turnaround time of the assays (68 minutes) in the materials and methods section. Furthermore, the rapidity of the assay was also clearly described in the discussion section of the manuscript (Line 362). This addition will ensure a comprehensive fulfilment of Objective 2 and provide a more robust analysis of our assay's performance.

We have also addressed the minor comments you provided:

• Line 32: Revised to "cefotaxime-supplemented agar plates."

• Lines 67-68: The accepted scientific notation for writing species names has been revised throughout the manuscript.

• Line 130: Since the AriaMx Real-Time PCR System is a fully integrated qPCR amplification, detection, and data analysis system, we did not see the need to include this description in the methods. 

• Line 219: A reference supporting the assertion that blaSHV is intrinsic to K. pneumoniae was included in the manuscript (Line 408 - 410).

• Lines 336-338: This has been revised to eliminate redundancy and enhance clarity.

• Lines 419-420: We have addressed the incomplete sentence in the conclusion section.

Reviewer 4:

We appreciate your thorough review of our manuscript, your constructive comments are invaluable, and we are grateful for the insights you provided. Here are our responses to your specific comments:

• Cost comparison of multiplex qPCR vs Culture: We agree that providing a cost comparison between multiplex RT-PCR and culture methods in our African setting is a relevant aspect to address. However, our focus was limited to the rapidity and modification of the assay. In the future, we will consider the practicality and potential advantages of our multiplex RT-PCR approach, especially in regions where qPCR might face challenges.

• Italicization of Genus and Species Names in Supplementary Tables: We acknowledge the oversight in not italicizing genus and species names in S2 and 6 Tables. This has been rectified in the revised version of the manuscript.

• Representation of Common Genes and Human Isolates in States: We appreciate your suggestion for a more streamlined presentation of common genes across states, particularly in the context of human isolates. However, because of the limited number of isolates (Enterobacter cloacae and Klebsiella pneumoniae) this was not considered but we already have a manuscript published describing the variations of the AMR genes in Escherichia coli from the different states (Olorunleke SO, Kirchner M, Duggett N, AbuOun M, Okorie-Kanu OJ, Stevens K, et al. Molecular characterization of extended-spectrum cephalosporin-resistant Escherichia coli isolated from livestock and in-contact humans in Southeast Nigeria. Front Microbiol. 2022;0: 2801. doi: https://10.3389/FMICB.2022.937968 )

• Incomplete Sentence and Recasting of Lines (420-423): This has been revised.

• Attribution of Highest Prevalence in Abia (Lines 387-390): We understand your point about the attribution of the highest prevalence of blaTEM and blaSHV in Abia to factors beyond its commercial hub status. We have added a study limitation section to the manuscript to acknowledge certain limitations about the sample size and the restricted number of Klebsiella and Enterobacter spp tested.

---

## [Decision Letter · Decision Letter 1]

2 Jan 2024

Rapid detection and molecular epidemiology of β-lactamase producing Enterobacteriaceae isolated from food animals and in-contact humans in Nigeria.

PONE-D-23-21745R1

Dear Dr. Olorunleke,

We’re pleased to inform you that your manuscript has been judged scientifically suitable for publication and will be formally accepted for publication once it meets all outstanding technical requirements.

Kind regards,

Kwame Kumi Asare, Ph.D

Academic Editor

PLOS ONE

Additional Editor Comments (optional):

Reviewers' comments:

Reviewer's Responses to Questions

**Comments to the Author**

1. If the authors have adequately addressed your comments raised in a previous round of review and you feel that this manuscript is now acceptable for publication, you may indicate that here to bypass the “Comments to the Author” section, enter your conflict of interest statement in the “Confidential to Editor” section, and submit your "Accept" recommendation.

Reviewer #1: All comments have been addressed

Reviewer #2: All comments have been addressed

Reviewer #4: All comments have been addressed

2. Is the manuscript technically sound, and do the data support the conclusions?

Reviewer #1: Yes

Reviewer #2: Yes

Reviewer #4: Yes

3. Has the statistical analysis been performed appropriately and rigorously? 

Reviewer #1: Yes

Reviewer #2: Yes

Reviewer #4: Yes

4. Have the authors made all data underlying the findings in their manuscript fully available?

Reviewer #1: Yes

Reviewer #2: Yes

Reviewer #4: Yes

5. Is the manuscript presented in an intelligible fashion and written in standard English?

Reviewer #1: Yes

Reviewer #2: Yes

Reviewer #4: Yes

6. Review Comments to the Author

Reviewer #1: (No Response)

Reviewer #2: (No Response)

Reviewer #4: The Authors have addressed all my queries. The genus and species names have all been italicized in Tables 2 and 6 and the incomplete sentence has been completed and clearer now. Authors have added a section on limitations of the study in response to my query while the cost between qPCR and culture has been addressed

7. PLOS authors have the option to publish the peer review history of their article (what does this mean?). If published, this will include your full peer review and any attached files.

Reviewer #1: **Yes: **Babafela Awosile

Reviewer #2: **Yes: **Chukwu Emelda Eberechukwu

Reviewer #4: **Yes: **Prof. Stella I. Smith

---

## [Editor Report · Acceptance letter]

1 Apr 2024

PONE-D-23-21745R1 

PLOS ONE

Dear Dr. Olorunleke, 

I'm pleased to inform you that your manuscript has been deemed suitable for publication in PLOS ONE. Congratulations! Your manuscript is now being handed over to our production team.

Kind regards, 

on behalf of

Dr. Kwame Kumi Asare 

Academic Editor

PLOS ONE